# Associations between In-Hospital Mortality and Prescribed Parenteral Energy and Amino Acid Doses in Critically Ill Patients: A Retrospective Cohort Study Using a Medical Claims Database

**DOI:** 10.3390/nu16010057

**Published:** 2023-12-24

**Authors:** Hideto Yasuda, Yuri Horikoshi, Satoru Kamoshita, Akiyoshi Kuroda, Takashi Moriya

**Affiliations:** 1Emergency and Critical Care Medicine, Jichi Medical University Saitama Medical Center, 1-847 Amanuma-cho, Ohmiya-ku, Saitama 330-0834, Japan; tmoriya@jichi.ac.jp; 2Medical Affairs Department, Research and Development Center, Otsuka Pharmaceutical Factory, Inc., 2-9 Kandatsukasa-machi, Chiyoda-ku, Tokyo 101-0048, Japan; horikoshi.yuri@otsuka.jp (Y.H.); kamoshitas@otsuka.jp (S.K.); 3Research and Development Center, Otsuka Pharmaceutical Factory, Inc., 2-9 Kandatsukasa-machi, Chiyoda-ku, Tokyo 101-0048, Japan; kuroda.akiyoshi@otsuka.jp

**Keywords:** amino acids, energy, critical illness, parenteral nutrition, mortality

## Abstract

Some critically ill patients completely rely on parenteral nutrition (PN), which often cannot provide sufficient energy/amino acids. We investigated the relationship between PN doses of energy/amino acids and clinical outcomes in a retrospective cohort study using a medical claims database (≥10.5 years, from Japan, and involving 20,773 adult intensive care unit (ICU) patients on mechanical ventilation and exclusively receiving PN). Study patients: >70 years old, 63.0%; male, 63.3%; and BMI < 22.5, 56.3%. Initiation of PN: third day of ICU admission. PN duration: 12 days. In-hospital mortality: 42.5%. Patients were divided into nine subgroups based on combinations of the mean daily doses received during ICU days 4–7: (1) energy (very low <10 kcal/kg/day; low ≥10, <20; and moderate ≥20); (2) amino acids (very low <0.3 g/kg/day; low ≥0.3, <0.6; and moderate ≥0.6). For each subgroup, adjusted odds ratios (AORs) of in-hospital mortality with 95% confidence intervals (CIs) were calculated by regression analysis. The highest odds of mortality among the nine subgroups was in the moderate calorie/very low amino acid (AOR = 2.25, 95% CI 1.76–2.87) and moderate calorie/low amino acid (AOR = 1.68, 95% CI 1.36–2.08) subgroups, meaning a significant increase in the odds of mortality by between 68% and 125% when an amino acid dose of <0.6 g/kg/day was prescribed during ICU days 4–7, even when ≥20 kcal/kg/day of calories was prescribed. In conclusion, PN-dependent critically ill patients may have better outcomes including in-hospital mortality when ≥0.6 g/kg/day of amino acids is prescribed.

## 1. Introduction

In critically ill patients, nutritional deficiency can increase mortality [1,2], and nutritional excess can increase infection complications and kidney injury [3,4]. In nutritional guidelines for critically ill patients [5,6], enteral nutrition (EN) is generally preferred over parenteral nutrition (PN). However, some critically ill patients must rely solely on PN because they are unable to receive EN due to gastrointestinal dysfunction and/or circulatory system instability [5]. For example, in a Japanese real-world study involving 99,147 critically ill patients receiving mechanical ventilation, 25.8% of the patients received PN alone as the nutritional management on day 7 of their intensive care unit (ICU) admission [7].

That same study revealed that in those patients receiving only PN, the median doses of energy and amino acids on day 7 in the ICU were substandard, at only 10.7 kcal/kg and 0.32 g/kg, respectively [7]. In contrast, the recommended daily doses of energy and amino acids in guidelines are 20 kcal/kg and at least 1.2 g/kg, respectively [5,6,8]. However, the recommended doses of nutritional components in guidelines are based mainly on studies in which EN was the primary form of nutrition provided; thus, these doses have been extrapolated to PN. That may not be optimal because PN differs from EN in the efficiency of nutrient absorption. Also, the severity of the illnesses and comorbidities of patients who receive PN often differ from those of patients who receive EN.

As far as we know, there have been no studies investigating the effects of different parenteral energy and amino acid doses or combinations of them on clinical outcomes in critically ill patients who are receiving only PN because they cannot take food orally or receive EN. It would be important to know whether there is a specific relationship between these PN doses and clinical outcomes in critically ill patients who must rely solely on PN.

The aim of this study was to determine the relationship between parenteral energy and amino acid doses and combinations of them during days 4 through 7 of an ICU admission and clinical outcomes in a population of fasting and mechanically ventilated patients who were identified using a national medical claims database from Japan.

## 2. Materials and Methods

### 2.1. Data Source

This was a retrospective cohort study using a medical claims database provided by Medical Data Vision Co., Ltd. (MDV, Tokyo, Japan). This database covers approximately 33 million patients from 400 Japanese hospitals representing approximately 23% of the acute care hospitals operating in Japan as of October 2020. This database contains information including dates of hospital admission and discharge, patient characteristics and diagnoses at admission (i.e., age, sex, height, weight, primary diagnosis, comorbidities, activities of daily living, and consciousness level), medical treatments and drugs prescribed during hospitalization, and clinical outcomes at the time of discharge. Diagnoses were identified using the International Statistical Classification of Diseases (ICD) 10th revision (ICD-10) (Appendix A). Medical treatments were identified using Japan-specific codes appearing in medical claims (Appendix A). The database also enabled extraction of information about whether oral dietary intake or EN was received and how many intravenous solution products were prescribed to patients. Doses of parenteral energy, amino acids, and lipids were calculated using the brand names and compositions of the solution products prescribed.

### 2.2. Patient Population

This study included patients who were hospitalized between January 2010 and June 2020, were aged ≥18 years, were in an ICU for ≥4 days, had survived and stayed in the hospital for >7 days, were fasting (i.e., received neither oral food intake nor EN) for >7 days, and had received mechanical ventilation on at least the first, second, and/or third day of ICU admission (Figure 1). The inclusion criteria were determined based on previous global nutritional research [9]. For this study, day 1 was regarded as the day of ICU admission. Patients were excluded from the study if their height, weight, or prescribed parenteral energy data were missing or suspected to be the result of input errors (i.e., height < 100 cm or ≥200 cm, weight < 10 kg or ≥200 kg, and/or energy prescribed of 0 kcal throughout the period of days 1 to 7). Patients whose height and weight data were missing or suspected to be the result of input errors were excluded because these data were needed to calculate the prescribed nutrition component doses. Also, patients whose prescribed energy dose was 0 kcal/kg/day for 7 days were excluded because it was suspected that data regarding their oral and enteral nutrition were missing.

### 2.3. Comparisons

Groups of patients were compared using the mean daily parenteral energy and amino acid doses during ICU admission days 4 through 7 because the administration of nutrition is recommended to be gradually increased during this period [10]. In order to clarify the associations between these doses and clinical outcomes, 3 separate comparisons were performed, each of which used the entire study population, as follows:

#### 2.3.1. Comparison 1: Parenteral Energy Dose

Study patients were divided into 3 groups based on the mean daily parenteral energy doses received during days 4 through 7 as follows: very low calories (<10 kcal/kg/day), low calories (≥10 kcal/kg/day and <20 kcal/kg/day), and moderate calories (≥20 kcal/kg/day). These cut-off values were determined based on guidelines recommending at least 20 kcal/kg/day of energy [5].

#### 2.3.2. Comparison 2: Parenteral Amino Acid Dose

Study patients were divided into 3 groups based on the mean daily parenteral amino acid doses received during days 4 through 7 as follows: very low amino acids (<0.3 g/kg/day), low amino acids (≥0.3 g/kg/day and <0.6 g/kg/day), and moderate amino acids (≥0.6 g/kg/day). These cut-off values were determined based on real-world studies from Japan, Australia, and New Zealand describing prescribed amino acid doses of 0.3 or 0.6 g/kg/day [7,9].

#### 2.3.3. Comparison 3: Parenteral Energy and Amino Acid Combinations

Study patients were divided into 9 groups based on combinations of the mean daily parenteral energy and amino acid doses received during days 4 through 7 using the same dose cut-offs described above in Comparisons 1 and 2 as follows: very low calories/very low amino acids, low calories/very low amino acids, moderate calories/very low amino acids, very low calories/low amino acids, low calories/low amino acids, moderate calories/low amino acids, very low calories/moderate amino acids, low calories/moderate amino acids, and moderate calories/moderate amino acids.

### 2.4. Endpoints

The primary endpoint was in-hospital mortality, and the secondary endpoints (used only for Comparisons 1 and 2) were length of hospital stay (LOS) and hospital readmission. Only patients who survived and were discharged from the hospital were evaluated for these secondary endpoints. Hospital readmission was defined as being admitted within 30 days of discharge to the same hospital as the initial admission.

### 2.5. Variables

Study variables are listed in Table 1 and Appendix A and consisted of hospital admission data extracted from the database as follows: age, sex, height, weight, number of beds of admission hospital, calendar year of admission, primary diagnosis (using ICD-10 codes, Appendix A), activities of daily living (assessed after being critically ill using the Barthel Index (BI) [11]), consciousness level (evaluated using the Japan Coma Scale (JCS): JCS0, alert; JCS1, 1-digit code, not fully alert but awake without any stimuli; JCS2, 2-digit code, arousable with stimulation; or JCS3, 3-digit code, unarousable [Appendix A] [12]), and surgeries under general or lumbar spinal anesthesia occurring between the day of hospital admission and the day of ICU admission. Extracted variables also included any of the following drugs or medical treatments prescribed or received during days 1 through 7: catecholamines, transfusions (i.e., fresh frozen plasma, platelets, and/or red blood cells), albumin, renal replacement therapy, intra-aortic balloon pump, plasmapheresis, extracorporeal membrane oxygenation (ECMO), intervention by a nutritional support team (NST), rehabilitation (for cardiac macrovascular disease, cerebrovascular disease, disuse syndrome, locomotor disease, and/or respiratory disease), and/or feeding therapy. In addition, body mass index (BMI) was calculated from height and weight data; severity of comorbidities was determined using extracted data and the Charlson Comorbidity Index score in conjunction with the algorithm developed by Quan et al. [13]; and malnutrition was based on the Global Leadership Initiative on Malnutrition for Asian populations [14] and defined as either BMI < 18.5 in patients < 70 years old or BMI < 20 in patients ≥ 70 years old.

Daily prescribed parenteral doses of energy, amino acids, lipids, and carbohydrates were calculated using extracted data pertaining to the number of parenteral nutrition products prescribed and the composition of those products, and the doses were reported per kilogram (kg) of ideal body weight. During the calculations of energy and carbohydrate doses, amounts from glucose solutions for drug preparation, maintenance, and extracellular fluid replacement were taken into consideration. Also, propofol, an anesthetic agent containing lipid emulsion as a solvent, was included in the calculations of energy and lipid doses. For calorie calculations, 4 kcal/g was used for carbohydrates and amino acids, and 9 kcal/g was used for lipids. Amino acid calculations were based on the total free amino acids contained in the parenteral nutrition formulations. Ideal body weight was determined based on a BMI of 22 and the actual patient height. The reason why we used ideal body weight for the calculation is that the amount of nutrition administered might be overestimated if the actual body weight was used because many study patients supposedly have a low BMI [7]. After individual patient calculations were completed, the mean daily doses of parenteral energy, amino acids, and lipids during days 1 through 7 were calculated for each group. In addition, median daily doses of energy and amino acids for each group were calculated and graphed to demonstrate trends over the first 7 days in the ICU. Also, the first day when PN (amino acids or lipid emulsion) was initiated after ICU admission, the total number of days of PN during the in-hospital period, and the total number of days of fasting (with neither oral intake nor EN) during the in-hospital period are shown.

### 2.6. Statistical Methods

Summary statistics for categorical variables are expressed as frequencies and percentages, and those for continuous variables are expressed as medians, first quartiles (Q1), and third quartiles (Q3). Either the chi-square test (for categorical variables) or analysis of variance (for continuous variables) was used to compare groups for outcomes. The significance level was set at 5% (2-sided).

Associations between groups and outcomes were evaluated using multivariate logistic regression analysis or multiple regression as appropriate and reported as adjusted odds ratios (AORs) or regression coefficients along with 95% confidence intervals (CIs). In Comparison 1, regression analyses for outcomes were performed with the very low-calorie patients as the reference. In Comparison 2, regression analyses were performed with the very low-amino acid patients as the reference. For both Comparisons, the very low group was selected to be the reference group in order to investigate the associations between the higher prescribed nutrition doses and clinical outcomes. In Comparison 3, regression analyses for in-hospital mortality were performed using the low-calorie/moderate-amino acid patients as the reference because these doses were related to the lowest mortality levels according to a previous study [15].

Both types of regression analyses were performed with adjustment for confounding factors that were selected based on the opinions of ICU specialists and previous high-quality database studies [16,17,18]. For patients whose Barthel Index or JCS could not be determined because of missing data, these variables were characterized as not available (NA). Because missing data for variables included in both types of regression analyses were not imputed, only the variables for complete cases were included in these analyses. All analyses were performed using SAS release 9.4 (SAS Institute, Inc., Cary, NC, USA).

## 3. Results

Of the 2,550,780 hospitalized patients from January 2010 through June 2020, 23,406 met the inclusion criteria, 2633 were excluded, and 20,773 were included in this study (Figure 1). The major patient characteristics for each study group are shown in Table 1. Characteristics not included in Table 1 (beds in admission hospital, admission years, JCS, and surgery) are shown in Appendix A.

### 3.1. Comparison 1: Parenteral Energy Dose

Of the 20,773 patients, 10,384 (50.0%) were in the very low-calorie group, 7103 (34.2%) were in the low-calorie group, and 3286 (15.8%) were in the moderate-calorie group (Table 1). The median (Q1, Q3) of the mean parenteral energy dose for days 1 through 7 for each of these groups was 4.8 (3.1, 6.5), 11.6 (9.8, 13.9), and 19.7 (17.1, 23.1) kcal/kg, respectively (Table 1). The median (Q1, Q3) energy dose on day 7 for each of these groups was 5.0 (2.8, 8.2), 16.2 (12.9, 19.3), and 25.3 (21.8, 29.8) kcal/kg, respectively (Figure 2A). Changes in doses of amino acids, lipids, and carbohydrates are shown in Appendix A. The proportion of patients with in-hospital mortality was highest in the very low-calorie group (45.5%, *p* < 0.001) (Table 2). The adjusted odds of in-hospital mortality were significantly lower for the low-calorie group (AOR = 0.85, 95% CI 0.78–0.92) relative to the very low-calorie reference group (Table 2). No significant differences were observed in the adjusted odds of hospital readmission or LOS among the three energy groups.

### 3.2. Comparison 2: Parenteral Amino Acid Dose

Of the 20,773 patients, 10,908 (52.5%) were in the very low-amino acid group, 5836 (28.1%) were in the low-amino acid group, and 4029 (19.4%) were in the moderate-amino acid group (Table 1). The median (Q1, Q3) of the mean parenteral amino acid dose days 1 through 7 for each group was 0.00 (0.00, 0.11), 0.32 (0.25, 0.41), and 0.59 (0.48, 0.73) g/kg, respectively (Table 1). The median amino acid dose on day 7 for each group was 0.0 (0.0, 0.2), 0.5 (0.4, 0.6), and 0.8 (0.7, 1.0) g/kg, respectively (Figure 2B). Changes in doses of energy, lipids, and carbohydrates are shown in Appendix A. The proportion of patients with in-hospital mortality was highest in the very low-amino acid group (47.3%, *p* < 0.001) (Table 2). The adjusted odds of in-hospital mortality were significantly lower for the low-amino acid group (AOR = 0.76, 95% CI 0.70–0.82) and the moderate-amino acid group (AOR = 0.69, 95% CI 0.63–0.76) relative to the very low-amino acid reference group (Table 2). No significant differences were observed in the adjusted odds of hospital readmission or LOS among the three amino acid groups.

### 3.3. Comparison 3: Parenteral Energy and Amino Acid Combinations

The adjusted odds of in-hospital mortality were significantly higher for the moderate-calorie/very low-amino acid group (AOR = 2.25, 95% CI 1.76–2.87) and the moderate-calorie/low-amino acid group (AOR = 1.68, 95% CI 1.36–2.08) relative to the low-calorie/moderate-amino acid reference group (Figure 3). No significant in-hospital mortality differences were observed for the other moderate-amino acid groups: the very low-calorie/moderate-amino acid (AOR = 1.10, 95% CI 0.78–1.56), and moderate-calorie/moderate-amino acid (AOR = 1.16, 95% CI 0.99–1.35) groups. These results imply that a significant increase in the odds of in-hospital death of between 68% and 125% occurred when amino acid doses of <0.6 g/kg/day were prescribed, even when ≥ 20 kcal/kg/day of calories was also prescribed.

## 4. Discussion

We investigated the effects of prescribed doses of parenteral energy and amino acids during days 4–7 of ICU admission on clinical outcomes in fasting (≥7 days) patients receiving mechanical ventilation. Relative to the low-calorie/moderate-amino acid group, patients in the moderate-calorie/very low-amino acid and moderate-calorie/low-amino acid groups had significantly higher adjusted odds of in-hospital mortality, whereas patients in the very low-calorie/moderate-amino acid and moderate-calorie/moderate-amino acid groups had odds that did not differ significantly.

This is the first study to investigate associations between prescribed doses of energy and amino acids and clinical outcomes in patients exclusively receiving PN after ICU admission. In the nutritional guidelines for ICU patients [19], an appropriate initiation timing of PN has not been clarified. However, the latest large-scale study on practical nutritional management for ICU patients receiving artificial nutrition [20] reports that PN was initiated on the second to third day (38 h) of ICU admission, which was close to our study results of the third day of ICU admission. In addition, a multicenter observational study conducted recently [21] demonstrated the association between an increase in the amino acid dose and a decrease in mortality, although the relationship between the energy dose and mortality was small, which supports our study results.

Previous randomized clinical studies have suggested that early achievement of the goal energy should be avoided in the acute phase of ICU admission [3,22]. Also, observational studies have shown that more protein provision as compared with lower intake of protein is associated with mortality reductions [1,2,15,23]. However, there has been no unified recommendation on the optimal combination of energy and amino acids. We believe ours is the first large-scale study of the associations between energy and amino acid dose combinations and clinical outcomes in fasting ICU patients who needed PN.

The AORs of in-hospital mortality for the very low-calorie/moderate-amino acid and moderate-calorie/moderate amino acid groups were around one (vs the reference group). In addition, the AORs of in-hospital mortality were the highest of all nine combination groups for the moderate-calorie/very low-amino acid and moderate-calorie/low-amino acid groups. These results suggest that when amino acid doses are at least half of the guideline-recommended dose (i.e., 0.6 g/kg/day), the energy doses administered during PN do not significantly impact in-hospital mortality. These findings indicate the relative importance of providing adequate amino acid doses to ICU patients and are supported by a randomized study of the effects of administered energy on clinical outcomes in ICU patients [24]. In that study, all received similar, adequate doses of protein, but one group was given only 40–60% of the calculated caloric requirements, while the other group received 70–100% of those requirements; ultimately, no inter-group differences in mortality were identified.

In critically ill patients, cytokines and other hormones are released as part of an acute, systemic inflammatory reaction; in addition, mitochondrial dysfunction, insulin resistance, and disorders of lipid metabolism are observed [25]. Such physiological changes promote skeletal muscle protein degradation to provide energy, which leads to a loss of lean body mass (LBM) [25]. Loss of LBM is reported to cause an increase in infectious complications, decrease in muscle strength, and increase in mortality [26,27]. In our study of patients fasting for ≥7 days after ICU admission, 63% of the patients were ≥ 70 years old, and 20% of the patients had a BMI of < 18.5, suggesting that many of these patients were at high risk for malnutrition. The inadequate provision of amino acids to patients with such risk is likely to aggravate malnutrition, reduce immune function, and further decrease LBM, leading to a deterioration in clinical outcomes.

In this real-world data study, only 19% of the patients were in the moderate-amino acid group, having been prescribed ≥0.6 g/kg/day of amino acids in their PN. Possible reasons for this finding include low-level awareness about optimal nutritional treatment among Japanese physicians [28]; priority use in Japan of 2-in-1 PN formulations containing low concentrations (3%) of amino acids [7]; and few Japanese patients receiving customized PN prescriptions. However, the insufficient administration of nutrition in critically ill patients has been observed not only in Japan but also all over the world, a phenomenon known as the “knowledge-to-action” gap [9,29]. These observations, combined with our finding that in-hospital mortality was significantly higher in all groups receiving low amino acid doses in their PN, suggest that particular emphasis needs to be placed on prescribing sufficient amino acid doses for critically ill patients depending on PN because of oral food and EN intolerance.

A common bias encountered when investigating the association between a prescribed treatment and clinical outcomes is “confounding by indication”. In this study, it was expected that the more severe the clinical condition of the patient, the less likely adequate nutrition would be prescribed because of the priority of other medical treatments and interventions and the worse the clinical outcomes would be. Thus, an adjustment for disease severity can be important in studies like ours. However, common disease severity scores, such as the APACHE II [30] and SOFA scores [31], were not available in the database used for this study. To compensate for this, we selected a wide variety of clinical confounding factors related to disease severity and used in previous high-quality database studies [16,17,18], and we employed these as adjustment factors in the multivariate analyses. Using these adjustment factors, it was found that while in-hospital mortality was highest in the very low-amino acid group (Comparison 2), it was not highest in the very low-calorie group (Comparison 1) or the very low-calorie/very low-amino acid group (Comparison 3). These results suggest that adjusting the regression analyses in this study using a variety of clinical factors that were potentially confounding minimized the potential bias of “confounding by indication”.

This study has several other limitations. First, the data used in this study were obtained from a medical claims database, which can be prone to input errors, missing data, and inaccuracies. Missing data were encountered for the Barthel Index (14.5%) and the JCS (<0.01%), and these were treated as missing characteristics in the multivariate analyses. The database did not contain data on actual doses administered or even on the remaining volumes of infusion solution, which were unused and likely discarded. As a result, it was necessary to use prescribed doses. Also, the methods used to determine dose orders for individual patients and the reasons why PN was selected for individual patients were not included in the database. However, an international ICU nutritional study [32] showed that about 33% of patients receive PN without any reason or basis in Japan and the proportion of such patients is higher than that in other Asian countries or other countries worldwide (6% to 7%). As about 30% of study patients were malnourished patients, patients at risk of refeeding syndrome [33] might be included, although detailed information was not available. However, the median of the mean energy dose prescribed during the period of days 1 through 7 for all the study patients was as small as less than 10 kcal/kg; therefore, the possibility of development of refeeding syndrome was considered to be low. The second limitation is related to external validity. The patients in our study had been fasting for ≥7 days and receiving only PN. As such, their disease conditions and severity likely differed from those of patients who primarily receive EN (which may suggest less severe illness) during their ICU admission. Therefore, the study results may not be generalizable to critically ill patients receiving EN. In addition, our study results might not be generalized to ICU patients in the US and European countries. About 55% of patients in this study had a BMI less than the standard (<22.5), indicating a significant difference in physique between the study patients and common patients in the US and European countries [34]. Nevertheless, this study is meaningful because it showed the associations between an increase in the prescribed amino acid dose and in-hospital mortality in not only patients with a high BMI but also those with a BMI less than the standard [1].

## 5. Conclusions

The odds of in-hospital mortality were significantly higher in critically ill ICU patients who were mechanically ventilated and received only PN with an amino acid dose less than 0.6 g/kg/day during days 4–7 of ICU admission even when a standard dose of energy was prescribed. Close attention should be paid to the amino acid dose prescribed to critically ill patients who are depending solely on PN because of oral food and EN intolerance, as this group may be especially at risk for poor clinical outcomes if amino acid doses of 0.6 g/kg/day or more are not prescribed.

## Figures and Tables

**Figure 1 nutrients-16-00057-f001:**
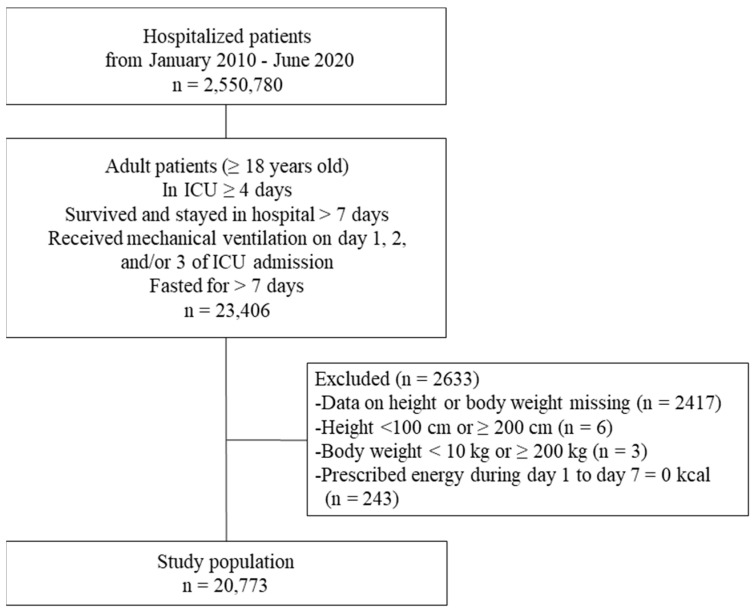
Study flow chart and disposition of 20,773 adult ICU patients hospitalized from January 2010 through June 2020 in Japan.

**Figure 2 nutrients-16-00057-f002:**
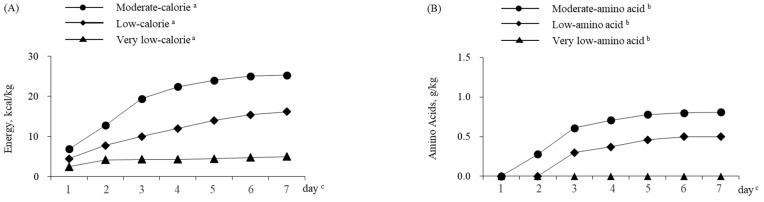
Change in median dose of energy in each group ^a^ (**A**) and change in median dose of amino acids in each group ^b^ (**B**) for 20,773 adult ICU patients hospitalized from January 2010 through June 2020 in Japan. Changes in median doses of lipids, carbohydrates, and other nutrients are shown in Appendix A. ^a^ Groups based on the mean daily parenteral energy dose days 4 through 7: very low calories (<10 kcal/kg/day); n = 10,384, low calories (≥10 and <20 kcal/kg/day): n = 7103, and moderate calories (≥20 kcal/kg/day): n = 3286. ^b^ Groups based on the mean daily parenteral amino acid dose days 4 through 7: very low amino acids (<0.3 g/kg/day); n = 10,908, low amino acids (≥0.3 and <0.6 g/kg/day); n = 5836, and moderate amino acids (≥0.6 g/kg/day); n = 4029. ^c^ Day 1 is regarded as the day of ICU admission.

**Figure 3 nutrients-16-00057-f003:**
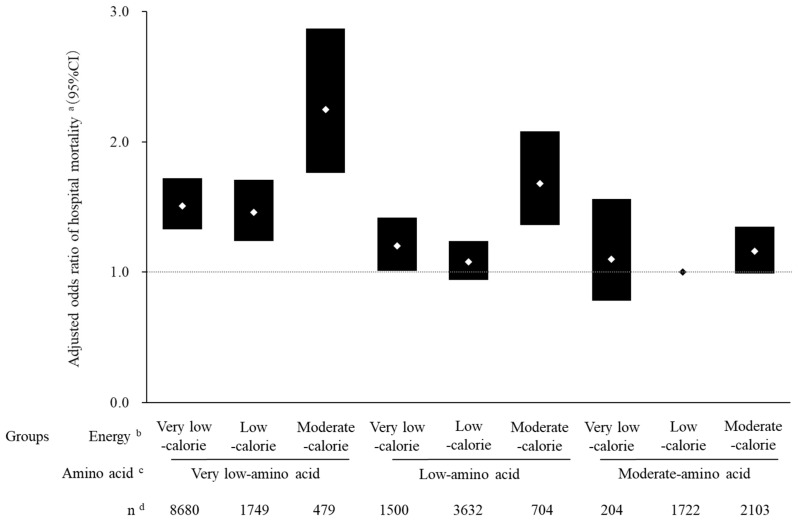
Adjusted ^a^ odds ratios of in-hospital mortality of 20,773 adult ICU patients hospitalized from January 2010 through June 2020 in Japan. Adjusted odds ratios (AORs) and 95% confidence intervals (CIs) were determined using multivariate logistic regression analysis, with the low-calorie/moderate-amino acid group as the reference. Each bar represents one of the nine groups, the AOR is denoted by diamond symbols, and the 95% CI is denoted by vertical lengths of bars. ^a^ Adjustments made for age, sex, BMI, beds in admission hospital, admission year, primary diagnosis, Charlson Comorbidity Index, Barthel Index, Japan Coma Scale, surgery, catecholamines, transfusion, albumin, renal replacement therapy, intra-aortic balloon pump, extracorporeal membrane oxygenation, nutritional support team, rehabilitation, and mean daily lipid doses during days 1 through 7. ^b^ Groups based on the mean daily parenteral energy dose days 4 through 7: very low calories (<10 kcal/kg/day), low calories (≥10 and <20 kcal/kg/day), and moderate calories (≥20 kcal/kg/day). ^c^ Groups based on the mean daily parenteral amino acid dose days 4 through 7: very low amino acids (<0.3 g/kg/day), low amino acids (≥0.3 and <0.6 g/kg/day), and moderate amino acids (≥0.6 g/kg/day). ^d^ Number of patients in each energy–amino acid combination group.

**Table 1 nutrients-16-00057-t001:** Characteristics of 20,773 adult ICU patients hospitalized from January 2010 through June 2020 in Japan.

Characteristics		Patient Groups Based on Energy Dose ^a^			Patient Groups Based on Amino Acid Dose ^b^		
		Very Low Calories	Low Calories	Moderate Calories	Very low Amino Acids	Low Amino Acids	Moderate Amino Acids
		n = 10,384	n = 7103	n = 3286	n = 10,908	n = 5836	n = 4029
Age, years, n (%)	<60	1926 (18.5)	1211 (17.0)	500 (15.2)	2056 (18.8)	954 (16.3)	627 (15.6)
	60–69	2000 (19.3)	1396 (19.7)	653 (19.9)	2148 (19.7)	1129 (19.3)	772 (19.2)
	70–79	2959 (28.5)	2211 (31.1)	1083 (33.0)	3194 (29.3)	1780 (30.5)	1279 (31.7)
	80–89	2898 (27.9)	1945 (27.4)	899 (27.4)	2948 (27.0)	1650 (28.3)	1144 (28.4)
	≥90	601 (5.8)	340 (4.8)	151 (4.6)	562 (5.2)	323 (5.5)	207 (5.1)
Sex, n (%)	Male	6537 (63.0)	4686 (66.0)	1929 (58.7)	6999 (64.2)	3854 (66.0)	2299 (57.1)
	Female	3847 (37.0)	2417 (34.0)	1357 (41.3)	3909 (35.8)	1982 (34.0)	1730 (42.9)
Body mass index ^c^, kg/m^2^, n (%)	<16	653 (6.3)	456 (6.4)	210 (6.4)	619 (5.7)	388 (6.6)	312 (7.7)
	16–<18.5	1432 (13.8)	979 (13.8)	482 (14.7)	1400 (12.8)	876 (15.0)	617 (15.3)
	18.5–<22.5	3615 (34.8)	2641 (37.2)	1222 (37.2)	3797 (34.8)	2138 (36.6)	1543 (38.3)
	22.5–<25	2149 (20.7)	1474 (20.8)	658 (20.0)	2301 (21.1)	1221 (20.9)	759 (18.8)
	≥25	2535 (24.4)	1553 (21.9)	714 (21.7)	2791 (25.6)	1213 (20.8)	798 (19.8)
Primary diagnosis ^d^, n (%)	Sepsis	451 (4.3)	529 (7.4)	271 (8.2)	605 (5.5)	412 (7.1)	234 (5.8)
	Neoplasm	645 (6.2)	988 (13.9)	639 (19.4)	727 (6.7)	700 (12.0)	845 (21.0)
	Diseases of the nervous system	517 (5.0)	194(2.7)	59 (1.8)	468 (4.3)	189 (3.2)	113 (2.8)
	Ischemic heart disease	738 (7.1)	441(6.2)	172 (5.2)	866 (7.9)	397 (6.8)	88 (2.2)
	Heart failure	666 (6.4)	330 (4.6)	144 (4.4)	719 (6.6)	320 (5.5)	101 (2.5)
	Cerebrovascular disorders	1617 (15.6)	570 (8.0)	224 (6.8)	1449 (13.3)	584 (10.0)	378 (9.4)
	Other circulatory system diseases	1837 (17.7)	1028 (14.5)	402 (12.2)	2148 (19.7)	733 (12.6)	386 (9.6)
	Pneumonia	474 (4.6)	247 (3.5)	85 (2.6)	448 (4.1)	243 (4.2)	115 (2.9)
	Interstitial respiratory diseases	299 (2.9)	227 (3.2)	72 (2.2)	289 (2.6)	209 (3.6)	100 (2.5)
	Other respiratory diseases	766 (7.4)	406 (5.7)	137 (4.2)	638 (5.8)	422 (7.2)	249 (6.2)
	Diseases of the digestive system	1158 (11.2)	1289 (18.1)	654 (19.9)	1248 (11.4)	938 (16.1)	915 (22.7)
	Kidney diseases	113 (1.1)	85 (1.2)	73 (2.2)	187 (1.7)	57 (1.0)	27 (0.7)
	Injury, poisoning, other consequences external causes	557 (5.4)	298 (4.2)	91 (2.8)	519 (4.8)	254 (4.4)	173 (4.3)
	Other	546 (5.3)	471 (6.6)	263 (8.0)	597 (5.5)	378 (6.5)	305 (7.6)
Charlson Comorbidity Index, n (%)	0	6075 (58.5)	3454 (48.6)	1456 (44.3)	6070 (55.6)	2936 (50.3)	1979 (49.1)
	1–2	3205 (30.9)	2572 (36.2)	1257 (38.3)	3496 (32.0)	2090 (35.8)	1448 (35.9)
	≥3	1104 (10.6)	1077 (15.2)	573(17.4)	1342 (12.3)	810 (13.9)	602 (14.9)
Barthel Index, n (%)	100	1582 (15.2)	1652 (23.3)	989 (30.1)	1877 (17.2)	1236 (21.2)	1110 (27.6)
	65–95	268 (2.6)	271 (3.8)	149 (4.5)	321 (2.9)	203 (3.5)	164 (4.1)
	45–60	278 (2.7)	252 (3.5)	119 (3.6)	285 (2.6)	221 (3.8)	143 (3.5)
	25–40	182 (1.8)	145 (2.0)	88 (2.7)	195 (1.8)	117 (2.0)	103 (2.6)
	5–20	422 (4.1)	334 (4.7)	150 (4.6)	420 (3.9)	282 (4.8)	204 (5.1)
	0	6166 (59.4)	3408 (48.0)	1308 (39.8)	6261 (57.4)	2912 (49.9)	1709 (42.4)
	NA	1486 (14.3)	1041 (14.7)	483 (14.7)	1549 (14.2)	865 (14.8)	596 (14.8)
Malnutrition ^e^, n (%)	<70 years and BMI < 18.5	538 (5.2)	458 (6.4)	207 (6.3)	568 (5.2)	368 (6.3)	267 (6.6)
	≥70 years and BMI < 20	2400 (23.1)	1622 (22.8)	763 (23.2)	2349 (21.5)	1403 (24.0)	1033 (25.6)
Prescription/Treatment ^f^, n (%)	Catecholamines	6971 (67.1)	5470 (77.0)	2608 (79.4)	7845 (71.9)	4252 (72.9)	2952 (73.3)
	Transfusions ^g^	4275 (41.2)	4010 (56.5)	2101 (63.9)	5206 (47.7)	2975 (51.0)	2205 (54.7)
	Albumin	3708 (35.7)	4110 (57.9)	2175 (66.2)	4612 (42.3)	3016 (51.7)	2365 (58.7)
	Renal replacement therapy	1800 (17.3)	1756 (24.7)	988 (30.1)	2704 (24.8)	1211 (20.8)	629 (15.6)
	Intra-aortic balloon pump	876 (8.4)	556 (7.8)	234 (7.1)	1082 (9.9)	474 (8.1)	110 (2.7)
	Plasmapheresis	39 (0.4)	31 (0.4)	36 (1.1)	56 (0.5)	28 (0.5)	22 (0.5)
	ECMO	489 (4.7)	302 (4.3)	107 (3.3)	639 (5.9)	203 (3.5)	56 (1.4)
	Nutritional support team	128 (1.2)	184 (2.6)	94 (2.9)	152 (1.4)	143 (2.5)	111 (2.8)
	Rehabilitation ^h^	4040 (38.9)	3328 (46.9)	1545 (47.0)	4014 (36.8)	2790 (47.8)	2109 (52.3)
Energy ^i^, kcal/kg/d, median [Q1, Q3]		4.8 [3.1, 6.5]	11.6 [9.8, 13.9]	19.7 [17.1, 23.1]	5.1 [3.2, 7.9]	10.7 [8.1, 13.9]	16.3 [12.6, 20.2]
Amino acids ^i^, g/kg/d, median [Q1, Q3]		0.00 [0.00, 0.15]	0.32 [0.20, 0.46]	0.52 [0.35, 0.69]	0.00 [0.00, 0.11]	0.32 [0.25, 0.41]	0.59 [0.48, 0.73]
Lipids ^i^, g/kg/d, median [Q1, Q3]		0.00 [0.00, 0.06]	0.04 [0.00, 0.13]	0.08 [0.01, 0.22]	0.01 [0.00, 0.09]	0.03 [0.00, 0.12]	0.04 [0.00, 0.15]
Carbohydrates, g/kg/d, median [Q1, Q3]		1.00 [0.67, 1.35]	2.37 [1.94, 2.89]	4.09 [3.49, 4.84]	1.09 [0.69, 1.68]	2.18 [1.59, 2.88]	3.28 [2.40, 4.10]
PN ^j^ initiation day, median [Q1, Q3]		6 [3, 9]	3 [2, 5]	2 [1, 3]	6 [3, 9]	3 [2, 4]	2 [1, 3]
PN ^j^ duration, days, median [Q1, Q3]		6 [0, 17]	17 [10, 31]	20 [12, 37]	5 [0, 17]	17 [10, 31]	20 [12, 36]
Fasting duration ^k^, days, median [Q1, Q3]		12 [9, 18]	13 [9, 21]	14 [9, 23]	12 [9, 19]	13 [9, 21]	13 [9, 22]

^a^ Groups based on the mean daily parenteral energy dose days 4 through 7: very low calories (<10 kcal/kg/day), low calories (≥10 and <20 kcal/kg/day), and moderate calories (≥20 kcal/kg/day). ^b^ Groups based on the mean daily amino acid dose days 4 through 7: very low amino acids (<0.3 g/kg/day), low amino acids (≥0.3 and <0.6 g/kg/day), and moderate amino acids (≥0.6 g/kg/day). ^c^ The number (proportion) of patients with BMI > 30 in each group was as follows: very low calories 633 (6.1%), low calories 377 (5.3%), moderate calories 155 (4.7%), very low amino acids 708 (6.5%), low amino acids 273 (4.7%), and moderate amino acids 184 (4.6%). ^d^ Primary diagnoses based on International Statistical Classification of Diseases, 10th revision (ICD-10) codes. ^e^ Malnutrition defined as either <70 years old and BMI <18.5 or ≥ 70 years old and BMI <20. ^f^ Prescriptions or treatments, based on Japan-specific codes, during days 1 through 7 of hospitalization. ^g^ Includes fresh frozen plasma, platelets, or red blood cells. ^h^ Rehabilitation for feeding therapy and/or one of the following types of diseases: cardiac microvascular, cerebrovascular, disuse syndrome, locomotor, and/or respiratory. ^i^ Median based on the mean daily doses during days 1 through 7 of hospitalization for each patient. ^j^ Administration of amino acids or lipid emulsion. ^k^ Total fasting (with neither oral intake nor EN) days during hospitalization. Abbreviations: ICU, intensive care unit; NA, not available (i.e., data for calculation unavailable); ECMO, extracorporeal membrane oxygenation; Q1, first quartile; and Q3, third quartile.

**Table 2 nutrients-16-00057-t002:** Distribution and adjusted odds ratios of clinical outcomes of 20,773 adult ICU patients hospitalized from January 2010 through June 2020 in Japan.

	Patient Groups Based on Energy Dose ^a^				Patient Groups Based on Amino Acid Dose ^b^			
Clinical Outcomes	Very Low Calories	Low Calories	Moderate Calories	*p*-Value	Very Low Amino Acids	Low Amino Acids	Moderate Amino Acids	*p*-Value
	n = 10,384	n = 7103	n = 3286		n = 10,908	n = 5836	n = 4029	
In-hospital mortality, n (%)	4724 (45.5)	2713 (38.2)	1384 (42.1)	***p* < 0.001 ^e^**	5159 (47.3)	2272 (38.9)	1390 (34.5)	***p* < 0.001 ^e^**
Hospital readmission ^cd^ n (%)	240 (4.2)	187 (4.3)	84 (4.4)	0.95 ^e^	258 (4.5)	151 (4.2)	102 (3.9)	0.42 ^e^
Length of hospital stay ^d^, days, median [Q1, Q3]	46 [30, 70]	47 [31, 73]	49 [32, 77]	***p* < 0.001 ^e^**	47 [30, 72]	47 [31, 74]	45 [29, 70]	***p* = 0.003 ^e^**
	AOR/regression coefficient ^fg^ (95%CI)				AOR/regression coefficient ^fg^ (95%CI)			
In-hospital mortality	Reference	**0.85 (0.78–0.92)**	1.09 (0.96–1.24)		Reference	**0.76 (0.70–0.82)**	**0.69 (0.63–0.76)**	
Hospital readmission ^cd^	Reference	1.11 (0.85–1.44)	1.18 (0.80–1.74)		Reference	0.96 (0.76–1.21)	0.81 (0.61–1.07)	
Length of hospital stay ^d^	Reference	1.83 (−0.89–4.55)	0.76 (−3.03–4.55)		Reference	1.13 (−1.66–3.93)	−0.98 (−4.27–2.30)	

^a^ Groups based on the mean daily parenteral energy dose days 4 through 7: very low calories (<10 kcal/kg/day), low calories (≥10 and <20 kcal/kg/day), and moderate calories (≥20 kcal/kg/day). ^b^ Groups based on the mean daily amino acid dose days 4 through 7: very low amino acids (<0.3 g/kg/day), low amino acids (≥0.3 and <0.6 g/kg/day), and moderate amino acids (≥0.6 g/kg/day). ^c^ Hospital readmission defined as being admitted within 30 days of discharge to the same hospital as the initial admission. ^d^ Only patients who survived (n = 11,952) were included in the analyses for hospital readmission and length of hospital stay. ^e^ Based on analysis of variance. ^f^ Based on multivariate logistic regression analysis or multiple regression, as appropriate, with results presented as adjusted odds ratios (AORs) or regression coefficients and 95% confidence intervals (CIs). ^g^ Adjusted for age, sex, BMI, beds in admission hospital, admission year, primary diagnosis, Charlson Comorbidity Index, Barthel Index, Japan Coma Scale, surgery, catecholamines, transfusion, albumin, renal replacement therapy, intra-aortic balloon pump, extracorporeal membrane oxygenation, nutritional support team, rehabilitation, and mean daily lipid doses during days 1 through 7. For the energy group comparison, additional adjustment was made for the mean daily amino acid doses during days 1 through 7. For the amino acid group comparison, additional adjustment was made for the mean daily energy doses during days 1 through 7. Results in **bold** are statistically significant. Abbreviations: ICU, intensive care unit; Q1, first quartile; and Q3, third quartile.

## Data Availability

The data presented in this study are available on request from the corresponding author. The data are not publicly available due to restriction of license.

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
