# Peer review of "Associations between In-Hospital Mortality and Prescribed Parenteral Energy and Amino Acid Doses in Critically Ill Patients: A Retrospective Cohort Study Using a Medical Claims Database"

_nutrients, 2023, doi:10.3390/nu16010057_

Round 1

Reviewer 1 Report

Comments and Suggestions for Authors

It is needed some improvement in some areas to enhance the potential of the present manuscript.

- It is very important to state the circumstances of PN administration. For example, it is not clear for me (when I am reading the manuscript) that PN was initiated on day 4. If this is correct,  I am agree with the authors, but you should support why PN was initiated on day 4 (Please cite the following, which is in agreement with how PN is being performed in your study (doi: 10.1016/j.medin.2019.12.17). Advise: read carefully and compare with recent original contemporary studies published in Parenteral Nutrition in Nutrients journal (doi: 10.3390/nu15214665.) to enrich discussion section. 

- Please, enlarge the information within the Abstract as much as possible; please, explain and describe largely your results (e.g., total mortality and characteristics of the population). Describe also, as well as within the manuscript, for how long PN was administered and when it was initiated in the ICU.

- Please, consider that the present study is describing the use of PN during the acute-stable phase (from day 4 to 7). If you agree, state this in discussion section. 

- Methods: Please, very briefly describe why all your calculations are based on ideal body weight and why you did not use normal weight in normal BMI subgroups. Cite some national or international nutrition GPCs. 

- It is important to explain (especially during intro, methods and results) when PN was initiated, for how long was initiated and if you know the exact reason why PN was initiated (in limitations authors explain it was not registered). If this cannot be explained, it shoud be explained as limitations of the study. 

- The majority of the population includes normal BMI patients, which is not really similar to western-world societies. It would be interesting to discuss briefly this, because it has been claimed the importance of protein delivery in obese, but your study shows that protein is always imprtant regarless the BMI subgroup or the body composition. 

- Your findings are original in terms of PN. There are similar recent findings that may be cited in discussion section (Evaluation of Nutritional Practices in the Critical Care patient (The ENPIC study): Does nutrition really affect ICU mortality? Clin Nutr ESPEN2022 Feb:47:325-332). Please, highlight that your finding is on patients with parenteral nutrition ([...] first large-scale study on the associations between energy and amino acid dose combinations, and clinical outcomes, in fasting ICU patients that needed PN.[...]).

- In limitations section, authors state that the patients in your study had been fasting for ≥7 days and receiving only PN. This is probably related with the reason why PN was initiated and it should be highlighted that no attemp to introduce enteral nutrition was performed. 

Author Response

Response to the Comments and Suggestions from Reviewer #1

  1. It is very important to state the circumstances of PN administration. For example, it is not clear for me (when I am reading the manuscript) that PN was initiated on day 4. If this is correct, I am agree with the authors, but you should support why PN was initiated on day 4 (Please cite the following, which is in agreement with how PN is being performed in your study (doi: 10.1016/j.medin.2019.12.17). Advise: read carefully and compare with recent original contemporary studies published in Parenteral Nutrition in Nutrients journal (doi: 10.3390/nu15214665.) to enrich discussion section.

<Response>

Thank you for your comments and advise. As you stated, the PN initiation timing is very important information; therefore, we have revised Table 1 and Discussion as follows.

Table 1: We have added the median of PN (amino acids, lipid) initiation date.

Discussion: We have added the following paragraph to compare the PN initiation timing between this study and published study with referring guideline.

(Lines 366-372) This is the first study which investigated associations between prescribed doses of energy and amino acids and clinical outcomes in the patients exclusively receiving PN after ICU admission. In the nutritional guideline for ICU patients [19], an appropriate initiation timing of PN has not been clarified. However, the latest large-scale study on practical nutritional management for ICU patients receiving artificial nutrition [20] reports that PN was initiated the 4th day of ICU admission, which was close to our study results, the 3rd day of ICU admission.

Information regarding the rationale of PN initiation timing was not obtained from the database (medical claims database) used for this study. This is one of the study limitations and we have already discussed it as study limitations; however, we have added some information as follows.

 (Lines 441-446) Also, the methods used to determine dose orders for individual patients and the reasons why PN was selected for individual patients were not included in the database. However, an international nutritional study in ICU [32] shows that about 33% of patients receive PN without any reason or basis in Japan and the proportion of such patients is higher than that in other Asian countries or other world countries (6% to 7%).

  1. Please, enlarge the information within the Abstract as much as possible; please, explain and describe largely your results (e.g., total mortality and characteristics of the population). Describe also, as well as within the manuscript, for how long PN was administered and when it was initiated in the ICU.

<Response>

Thank you for your comments. We have added the following information in Abstract.

(Lines 19-20) Study patients: >70 years, 63.0%; male, 63.3%; BMI <22.5, 56.3%. Initiation of PN: 3rd day of ICU admission. PN duration: 12 days. In-hospital mortality: 42.5%. 

  1. Please, consider that the present study is describing the use of PN during the acute-stable phase (from day 4 to 7). If you agree, state this in discussion section.

<Response>

Thank you for your comments. We agree your opinion and consider that the information on the evaluation period of prescribed nutrition is very important. Therefore, we have revised Abstract, Discussion, and Conclusions as follows.

(Lines 21-22) …based on combinations of mean daily doses received during 4-7th days of ICU:

(Lines 27-28) …when an amino acid dose of < 0.6 g/kg/day was prescribed during 4-7th days of ICU,

(Linse 465-466) …when amino acid doses less than 0.6 g/kg/day were prescribed during 4-7th days of ICU admission,

Also, the reasons why we used the mean value from 4th to 7th day of ICU have been added as follows in Materials and Methods.

(Lines 123-124) …because the nutrition administration is recommended to be gradually increased during such period [10].

  1. Methods: Please, very briefly describe why all your calculations are based on ideal body weight and why you did not use normal weight in normal BMI subgroups. Cite some national or international nutrition GPCs.

<Response>

Thank you for your suggestions. We have added the reason why we used the ideal body weight for all calculations in Materials and Methods as follows:

(Lines 190-193) The reason why we used ideal body weight for calculation is that the nutrition administration amount might be overestimated if the actual body weight was used because many of study patients are supposed to be with low BMI [7].

  1. It is important to explain (especially during intro, methods and results) when PN was initiated, for how long was initiated and if you know the exact reason why PN was initiated (in limitations authors explain it was not registered). If this cannot be explained, it should be explained as limitations of the study.

<Response>

Thank you for your comments and suggestions. As you stated, the PN initiation timing and PN duration period are very important information; therefore, we have revised Table 1 as follows.

Table 1: We have added the PN initiation day and the median of PN duration period.

As we have already explained in study limitations, reasons why PN was selected were not included in the database. Therefore, we have added the results of another study on clinical practice just for reference, as follows.

(Lines 443-446) However, an international nutritional study in ICU [32] shows that about 33% of patients receive PN without any reason or basis in Japan and the proportion is higher than that in other Asian countries or other world countries (6% to 7%).

  1. The majority of the population includes normal BMI patients, which is not really similar to western-world societies. It would be interesting to discuss briefly this, because it has been claimed the importance of protein delivery in obese, but your study shows that protein is always important regardless the BMI subgroup or the body composition.

<Response>

Thank you for your comments. As your opinion is considered to be the issue related to the external validity, we have added the following sentences in study limitations.

(Lines 455-462) In addition, our study results might not be generalized to ICU patients in the US and European countries. About 55% of patients in this study had BMI less than standard (<22.5) meaning that a significant difference in physique between the study patients and common patients in the US and European countries [34]. Nevertheless, this study is meaningful because it showed the associations between increase of prescribed amino acid dose and in-hospital mortality existed in the patients many of who had BMI less than standard, not only in the patients with high BMI [35].

  1. Your findings are original in terms of PN. There are similar recent findings that may be cited in discussion section (Evaluation of Nutritional Practices in the Critical Care patient (The ENPIC study): Does nutrition really affect ICU mortality? Clin Nutr ESPEN. 2022 Feb:47:325-332). Please, highlight that your finding is on patients with parenteral nutrition ([...] first large-scale study on the associations between energy and amino acid dose combinations, and clinical outcomes, in fasting ICU patients that needed PN.[...]).

<Response>

Thank you for your comments and suggestions. As you stated, we should highlight that our study is novel because it is the first one which investigated the associations between energy and amino acid doses and clinical outcomes in ICU patient exclusively receiving PN. Therefore, we had prepared a new paragraph to discuss it in Discussion. Also, we have referred the ENPIC study as the paper which supports our study in the same paragraph.

(Lines 366-375) This is the first study which investigated associations between prescribed doses of energy and amino acids and clinical outcomes in the patients exclusively receiving PN after ICU admission. …In addition, a multicenter observational study conducted recently [21] demonstrates the association between increase of amino acid dose and decrease of mortality, although the relationship between energy dose and mortality was small, which supports our study results.

In addition, the following sentence has been revised according to your suggestions.

(Line 382) …and clinical outcomes, in fasting ICU patients that needed PN.

  1. In limitations section, authors state that the patients in your study had been fasting for ≥7 days and receiving only PN. This is probably related with the reason why PN was initiated and it should be highlighted that no attempt to introduce enteral nutrition was performed.

<Response>

Thank you for your thoughtful opinions. As we answered to your comment No.5, we could not obtain the information on the reasons why PN was selected; therefore, we consider that it is one of study limitations. As the possibility of PN selection without any reason or basis might be higher in Japan than in other countries, we had added the following sentence in study limitations just for reference.

(Lines 443-446) However, an international nutritional study in ICU [32] shows that about 33% of patients receive PN without any reason or basis in Japan and the proportion is higher than that in other Asian countries or other world countries (6% to 7%).

Reviewer 2 Report

Comments and Suggestions for Authors

This manuscript describes a retrospective cohort study conducted in Japan, spanning over 10 years and involving over 20,000 adult intensive care unit patients who were on mechanical ventilation and exclusively receiving parenteral nutrition. The study aimed to investigate the relationship between the doses of energy and amino acids administered through PN and the clinical outcomes of these critically ill patients. The study is interesting and has an impressive number of patients included for an interesting research question.

*Table 1 is interesting but difficult to read because of it's size. Still ,the authors have not included the carbohydrate count even if they comment that data is available (including dextrose contribution from drug infusions). I suggest Table 1 be adjusted: i) reduced in size by moving less interesting info to Supplement section and ii) include also the carbohydrate contribution. 

*The data in fig2 ,you can see that aa=0 for one group of patients however they get some energy. Does this come from lipids/propofol or dextrose prescribed as PN or from drug infusions? Just including lipids and carbohydrates in a graph would be nice to see. 

*The figure text of fig3 is too long ,superscripts are missing , the manuscript becomes difficult to read with so extreme table/figure legends in relation to the results section text.  

* Reefeding syndrom is not mentioned at all. I believe this is highly relevant to discuss in the studied cohort, see eg Curr Opin Crit Care 2018, 24:235–240 DOI:10.1097/MCC.0000000000000514 

* in section 2.5 lumbar is amusingly misspelled.

Author Response

Response to the Comments and Suggestions from Reviewer #2

  1. Table 1 is interesting but difficult to read because of it's size. Still, the authors have not included the carbohydrate count even if they comment that data is available (including dextrose contribution from drug infusions). I suggest Table 1 be adjusted: i) reduced in size by moving less interesting info to Supplement section and ii) include also the carbohydrate contribution.

<Response>

Thank you for your thoughtful opinions. We had transferred the number of beds in admission hospital, admission years, JCS, and surgery from Table 1 to Additional file 3. In addition, median of the mean carbohydrate dose prescribed during the period of days 1 to 7 has been added in Table 1.

  1. The data in fig2, you can see that aa=0 for one group of patients however they get some energy. Does this come from lipids/propofol or dextrose prescribed as PN or from drug infusions? Just including lipids and carbohydrates in a graph would be nice to see.

<Response>

Thank you for your suggestions. We have added figures of the change in prescribed doses of amino acids, lipid, and carbohydrate for each study group in Additional file 4.

  1. The figure text of fig3 is too long, superscripts are missing, the manuscript becomes difficult to read with so extreme table/figure legends in relation to the results section text.

<Response>

Thank you for your comments. Primary results of ORs have already shown in Results; therefore, we have deleted such information from the footnote of Fig. 3. Also, we have corrected the superscripts as you stated.

  1. Refeeding syndrome is not mentioned at all. I believe this is highly relevant to discuss in the studied cohort, see eg Curr Opin Crit Care 2018, 24:235–240 DOI:10.1097/MCC.0000000000000514.

<Response>

Thank you for your thoughtful comments. We also consider that refeeding syndrome should be mentioned in this field; however, we could not obtain detail information on the nutritional conditions of patients from the database used for this study. As this is one of the study limitations, we had added the following sentences in study limitations.

(Lines 446-451) As about 30% of study patients were malnourished patients, patients at the risk of refeeding syndrome [33] might be included, although detail information was not available. However, the median of the mean energy dose prescribed during the period of days 1 to 7 for whole study patients was as small as less than 10 kcal/kg (data not shown); therefore, the possibility of development of refeeding syndrome is considered to be low.

  1. In section 2.5 lumbar is amusingly misspelled

<Response>

Thank you for your comments. We have corrected the misspelling.

(Line 166) …and surgeries under general or lumbar spinal anesthesia occurring…

Round 2

Reviewer 2 Report

Comments and Suggestions for Authors

N/A